# C-Reactive Protein-to-Albumin Ratio to Predict Tolerability of S-1 as an Adjuvant Chemotherapy in Pancreatic Cancer

**DOI:** 10.3390/cancers16050922

**Published:** 2024-02-25

**Authors:** Naotake Funamizu, Akimasa Sakamoto, Takahiro Hikida, Chihiro Ito, Mikiya Shine, Yusuke Nishi, Mio Uraoka, Tomoyuki Nagaoka, Masahiko Honjo, Kei Tamura, Katsunori Sakamoto, Kohei Ogawa, Yasutsugu Takada

**Affiliations:** Department of Hepato-Biliary-Pancreatic Surgery, Ehime University Graduate School of Medicine, Toon 791-0295, Japan; sakamoto.akimasa.kw@ehime-u.ac.jp (A.S.); hikida.takahiro.ny@ehime-u.ac.jp (T.H.); chippy.ito@gmail.com (C.I.); shine.mikiya.kz@ehime-u.ac.jp (M.S.); nishi.yusuke.jp@ehime-u.ac.jp (Y.N.); uraoka.mio.lr@ehime-u.ac.jp (M.U.); muscle_tomoyuki@yahoo.co.jp (T.N.); honjo.masahiko.ce@ehime-u.ac.jp (M.H.); k-tamura@m.ehime-u.ac.jp (K.T.); sakamoto.katsunori.gq@ehime-u.ac.jp (K.S.); ogawa.kohei.vz@ehime-u.ac.jp (K.O.); takaday@m.ehime-u.ac.jp (Y.T.)

**Keywords:** adjuvant chemotherapy, C-reactive protein-to-albumin ratio, malnutrition, pancreatic cancer, S-1

## Abstract

**Simple Summary:**

Adjuvant chemotherapy (AC) with S1 is beneficial for pancreatic cancer, but the completion rate of S1 remains at 70%. Additionally, there are no useful indicators for achieving completion of S1. Therefore, we have assumed that studying the C-reactive protein-to-albumin ratio as an indicator based on nutritional status may be useful in predicting this. When this indicator proves effective, we believe that more cases can complete the treatment by improving nutrition or adjusting the dosage before starting AC.

**Abstract:**

Adjuvant chemotherapy (AC) with S-1 after radical surgery for resectable pancreatic cancer (PC) has shown a significant survival advantage over surgery alone. Consequently, ensuring that patients receive a consistent, uninterrupted S-1 regimen is of paramount importance. This study aimed to investigate whether the C-reactive protein-to-albumin ratio (CAR) could predict S-1 AC completion in PC patients without dropout due to adverse events (AEs). We retrospectively enrolled 95 patients who underwent radical pancreatectomy and S-1 AC for PC between January 2010 and December 2022. A statistical analysis was conducted to explore the correlation of predictive markers with S-1 completion, defined as continuous oral administration for 6 months. Among the 95 enrolled patients, 66 (69.5%) completed S-1, and 29 (30.5%) failed. Receiver operating characteristic curve analysis revealed 0.05 as the optimal CAR threshold to predict S-1 completion. Univariate and multivariate analyses further validated that a CAR ≥ 0.05 was independently correlated with S-1 completion (*p* < 0.001 and *p* = 0.006, respectively). Furthermore, a significant association was established between a higher CAR at initiation of oral administration and acceptable recurrence-free and overall survival (*p* = 0.003 and *p* < 0.001, respectively). CAR ≥ 0.05 serves as a predictive marker for difficulty in completing S-1 treatment as AC for PC due to AEs.

## 1. Introduction

Pancreatic cancer (PC) is a highly malignant tumor with low morbidity. Surgical resection is necessary to achieve a cure in PC. However, despite advancements in surgical techniques and perioperative management, the 5-year survival rate remains low, at only 12%. This is primarily because >90% of patients experience local recurrence or distant metastasis following radical resection [1]. Several clinical trials have yielded compelling evidence that in cases of resectable PC, combined radical resection and adjuvant chemotherapy (AC) results in a significantly improved prognosis compared to surgery alone [2,3,4]. Notably, the Japanese guidelines recommend the oral administration of S-1 as the standard AC protocol following radical surgery for PC patients with PC [5]. This recommendation is based on the findings of a randomized trial conducted by the Japan Adjuvant Study Group of Pancreatic Cancer (JASPAC01), which demonstrated that patients who received S-1 exhibited significantly longer 5-year and median survival rates than those who were administered gemcitabine following radical surgery [6]. Despite the established efficacy of AC in improving the prognosis of PC patients, achieving successful completion can be challenging because of the occurrence of postoperative complications (POCs) or adverse events (AEs) associated with the AC regimen itself [7]. However, the reported completion rates for AC due to S-1 have been suboptimal, typically falling by approximately 70% [8,9]. Discontinuation of AC regimens can compromise their anticancer effectiveness, resulting in adverse implications on patient prognosis. Hence, clinicians urgently need to identify reliable indicators for predicting the successful completion of S-1 treatment. Recent reports have suggested a potential correlation between nutritional status and treatment completion rates for AC [10,11,12]. Consequently, the primary objective of this study was to evaluate whether the C-reactive protein (CRP)-to-albumin ratio (CAR), a recognized nutritional marker, could act as a novel predictor for the rate of S-1 non-completion due to AEs in patients with PC.

## 2. Materials and Methods

In this retrospective study, we enrolled 186 consecutive patients who underwent radical pancreatic resection for PC at Ehime University Hospital (Toon City, Japan) between January 2010 and December 2022. Notably, 60 patients were excluded, as they did not initiate S-1 AC. Consequently, the final analysis included a cohort of 126 patients (Figure 1). It is worth highlighting that no patient mortality occurred before postoperative day 90. This study involved an extensive review of the patients’ medical records, encompassing the collection of data pertaining to patient backgrounds, perioperative laboratory results, perioperative clinical information, pathological findings, and postoperative prognoses.

During pancreatoduodenectomy, the majority of pancreatic anastomoses to the alimentary tract were carried out using the end-to-side pancreatojejunostomy technique, and as part of the routine procedure, two closed suction drainage tubes were inserted. In cases of distal pancreatectomy, pancreatic resection was predominantly accomplished using a linear stapler, and one or two closed suction drainage tubes were routinely placed, in accordance with the surgeon’s preference. To assess POCs, we applied the Clavien–Dindo (CD) classification, with grade ≥3 complications being categorized as major POCs [13].

At our hospital, AC was scheduled to commence promptly following hospital discharge and to continue for 6 months. Most patients initiated AC within 3 months of curative surgery. The therapeutic regimen included oral administration of S-1, ranging from 80 to 120 mg, contingent on body surface area, twice a day for 28 days, followed by a 14-day rest period. Treatment courses were repeated over 6 months unless intolerable toxicity occurred. Relative dose intensity (RDI) was calculated as the ratio of the actual dose intensity to the standard or planned S-1 dose intensity [14]. Completion of S-1 therapy was defined as persistent oral administration of S-1 with an RDI exceeding 80% [15,16,17]. Hematological and biochemical analyses, in conjunction with clinical parameters, including body weight fluctuations, were assessed at AC initiation and during all follow-up appointments. Postoperative surveillance was performed using contrast-enhanced CT on a monthly basis. AEs were evaluated in accordance with the Common Terminology Criteria for Adverse Events version 5.0, and AEs with grade ≥3 were categorized as severe AEs [16,18].

The CRP and albumin levels were measured on the same day, before the initiation of AC, and they were used to calculate the CAR. The timing of blood collection was determined by utilizing the blood test results taken when each surgeon considered it appropriate to commence adjuvant chemotherapy. CAR was calculated using the following formula: CAR = [CRP (mg/dL)]/[albumin (g/dL)] [17,19]. Following the determination of the CAR threshold value, patients were stratified into two groups: S-1-complete group and S-1-incomplete group.

All statistical analyses were conducted using the Statistical Package for the Social Sciences version 16.0 for Windows^®^ (IBM Corp., Armonk, NY, USA) and GraphPad Prism version 5.0 (GraphPad Software Inc., La Jolla, CA, USA). Patient demographics are presented as medians and interquartile ranges for nonparametric distributions, whereas categorical data are presented as numbers and percentages. The statistical significance of patient demographics and outcomes was assessed using the χ^2^ test, Fisher’s exact test, and the U test, as appropriate. Univariate and multivariate analyses were used to identify independent factors affecting the completion of S-1 administration. To determine the optimal cutoff value for CAR in predicting the risk of S-1 incompletion, a receiver operating characteristic (ROC) curve analysis was performed, and the cutoff value was determined using the Youden index. Overall survival (OS) and recurrence-free survival (RFS) following curative surgery were evaluated using the Kaplan–Meier method, and survival curves were compared using the log-rank test. A *p* value < 0.05 was considered statistically significant.

## 3. Results

### 3.1. Patients Characteristics with or without S-1 Completion

During the study period, 126 patients underwent curative surgery for PC, after which they were initiated on S-1 as the AC regimen. Among these patients, 66 (69.5%) continued AC, achieving an RDI > 80%. In contrast, 31 patients experienced recurrence during AC, prompting a modification of the treatment regimen. Moreover, 24 individuals required dose reduction and/or treatment interruption owing to AEs. Four patients terminated AC due to POCs, such as cholangitis, whereas one patient discontinued treatment due to an unrelated condition (cerebral infarction) (Figure 1). The detailed patient characteristics are presented in Table 1. Additionally, laboratory test findings at the onset of AC as well as relevant AC-related factors are outlined in Table 2.

There were no differences in the initiation period of AC and carbohydrate antigen 19-9 values between the S-1-complete and S-1-non-complete groups. However, significant differences were observed in the body mass index (BMI) and CAR between the groups (*p* = 0.026 and *p* < 0.001, respectively).

### 3.2. Calculation of the Optimal CAR

The optimal cutoff value was determined using ROC curve analysis (Figure 2). The areas under the ROC curves for the CAR, albumin, and CRP levels were 0.806, 0.704, and 0.799, respectively. The Youden index indicated that the most appropriate cutoff value for the CAR was 0.05, with a sensitivity of 72.7%, a specificity of 82.8%, and a likelihood ratio of 4.22. Patients were stratified into two groups based on the CAR cutoff value: the higher-CAR group (CAR ≥ 0.05, *n* = 42) and the lower-CAR group (CAR < 0.05, *n* = 53). Failure of S-1 completion occurred in 24 patients (57.1%) in the higher-CAR group and in only 5 patients (9.4%) in the lower-CAR group. Univariate analysis was conducted to assess whether a CAR value ≥ 0.05 can serve as a risk factor for the failure of S-1 treatment following surgery (*p* < 0.001) (Table 3).

### 3.3. Multivariate Analysis for S-1 Completion

All predictive factors that correlated with S-1 completion on univariate analysis were included in the multivariate analysis (Table 4). In the multivariate analysis, CAR < 0.05 and BMI were identified as independent risk factors for S-1 completion (hazard ratio [HR], 12.734; 95% confidence interval [CI], 2.064–79.253; *p* = 0.006; and HR, 1.322; 95% CI, 1.041–1.680; *p* = 0.022, respectively).

### 3.4. CAR and Outcome

The prognostic value of the CAR was also investigated (Figure 3). Patients with CAR ≥ 0.05 had worse RFS (HR, 0.323; 95% CI, 0.155–0.671; *p* = 0.002) and OS (HR, 0.249; 95% CI, 0.119–0.523; *p* = 0.001) than patients with CAR < 0.05.

## 4. Discussion

S-1 is a groundbreaking oral anticancer medication meticulously crafted from tegafur, a prodrug renowned for its transformative potential into three potent chemotherapeutic agents: fluorouracil, gimeracil, and potassium oteracils. Gimeracil’s mode of action is intricately tied to its ability to inhibit the activity of dihydropyrimidine dehydrogenase, a pivotal enzyme involved in the metabolism of fluorouracil. By impeding this enzyme, gimeracil effectively boosts the levels of fluorouracil both in the bloodstream and within tumor tissues, thereby enhancing its therapeutic efficacy. Conversely, oteracil potassium, another essential component of S-1, assumes a pivotal role in the complex interplay of pharmacodynamics. It exerts its influence by actively suppressing the phosphorylation of fluorouracil within the gastrointestinal tract. This mechanism significantly mitigates the risk of gastrointestinal toxicity associated with fluorouracil administration, ensuring a more tolerable treatment experience for patients undergoing chemotherapy. The clinical efficacy of S-1 has been extensively studied and validated, particularly in the context of adjuvant chemotherapy for various cancer types. Notably, in Japanese patients diagnosed with gastric cancer, as evidenced by multiple studies [8], S-1 has exhibited remarkable efficacy in conferring survival benefits and improving overall prognosis. Similarly, in the ASCOT trial involving patients with biliary tract cancer [9], S-1 has demonstrated compelling evidence of its ability to enhance patient outcomes when used as part of a comprehensive treatment regimen. The standard treatment for PC with AC was established based on the findings of the JASPAC01 trial [6]. Although tolerance of S-1 is generally considered higher in the Asian population than in the Caucasian population, it remains a challenge for all patients [20,21]. Fundamentally, in both of the two prior large-scale trials investigating S-1 (ASCOT [9] and JASPAC01 [6]), 72% of the patients successfully completed S-1 therapy; in this study, 69.5% of the patients completed AC treatment, which is consistent with earlier findings. Owing to the high rate of treatment failure, identification of a marker to predict the completion of S-1 treatment in a non-invasive and straightforward manner would be valuable for assessing patient prognosis and postoperative follow-up intervals.

In recent years, reports have suggested that nutritional status could influence POCs [22,23,24,25,26]. Additionally, the relationship between nutritional status and prognosis has been intensively investigated in a variety of cancer types, emphasizing the significance of assessing and improving patients’ nutritional status [27,28,29,30]. Conversely, some reports have suggested that inflammation can also affect POCs and prognosis [31,32,33]. Therefore, we assumed that indicators encompassing both the nutritional status and inflammatory conditions would be more promising. Various indicators, including the lymphocyte-to-CRP ratio [34], platelet-to-albumin ratio [35], neutrophil-to-lymphocyte ratio [36,37], platelet-to-lymphocyte ratio [38], prognostic nutritional index [39,40], and CAR [41,42], which take into account both inflammation and nutritional status, have been identified as predictors of prognosis or POCs. Among these indicators, the identification of one as a useful prognostic marker may also allow us to predict the S-1 completion rates for AC, which are correlated with better prognosis in PC patients. Therefore, in the present study, we focused on the CAR as an optimal novel index to predict S-1 completion, and thereby the prognosis of PC patients. The CAR has been reported to be associated with POCs and prognosis in patients with cancer or severe infections [40,43,44]. Moreover, CAR has been suggested to be associated with POCs and prognosis of PC [45,46]. Various nutritional indicators have been studied in relation to AC tolerance due to S-1 in several types of cancers, including body weight loss [47], albumin-bilirubin score, geriatric nutritional risk index, prognostic nutritional index, and neutrophil-to-lymphocyte ratio [12,48,49].

In this study, CAR was higher in the S-1-incomplete group than in the S-1-complete group. An observed CAR value of 0.05 or greater was linked to an increased risk of S-1 therapy incompletion due to AEs, indicating that enhancing prechemotherapeutic nutritional or inflammatory status could potentially reduce the incidence of failure of S-1 treatment caused by AEs. Furthermore, a CAR value of 0.05 or higher not only serves as an indicator of the successful completion of S-1 therapy but also holds promise as a significant prognostic marker. This observation, however, is not unexpected, considering the well-established improvement in overall prognosis associated with AC utilizing S-1. Additionally, the consistency of CAR values equal to or greater than 0.05, with the CAR value previously identified as a predictive factor for POCs in a prior study involving PC patients from our institution [42], provides further validation of its prognostic relevance in the context of PC treatment.

According to a large-scale study, patients who experienced severe POCs, specifically those with a CD grade ≥3, were reported to initiate AC less frequently after pancreatic surgery [50]. Furthermore, it has been demonstrated that CAR serves as a valuable predictor of POCs following pancreatic resection [41,42]. CAR, a straightforward metric calculated solely through blood tests, has been underscored as a prognostic factor for PC patients. This result suggests the increasing importance of managing the nutritional status of patients with cancer, in addition to monitoring CAR. As mentioned above, numerous nutritional assessment indices have been documented as prognostic indicators of PC. Notably, CAR has been independently reported as a prognostic factor of PC in numerous studies [46,51,52]. Thus, our results are consistent with these findings. Furthermore, body weight loss has been reported to hinder S-1 completion [47,53], and our data indicate that a lower BMI is a predictive factor of treatment failure. Accordingly, the failure to recover postoperative weight loss prior to administering AC with S-1 suggests a heightened risk of dropout due to AEs. Given the close association between weight loss and nutritional status, this finding aligns with our data, indicating S-1 incompletion in cases with elevated CAR values.

However, this study has some limitations. First, this is a single-center study with a relatively small dataset, which may have introduced bias into the data analysis and thereby potentially limited our ability to fully assess the impact of the CAR. Second, the retrospective nature of this study may have introduced selection bias. Finally, the details of S-1 treatment, including the timing of initiation, dosage, dose reduction, and withdrawal, were determined by the physician. Moreover, in this study, an RDI > 80 was defined as S-1 completion, but the appropriateness of the cutoff value of 80 has not been conclusively determined. Therefore, the results of this study should be verified in a large-scale series.

## 5. Conclusions

In this groundbreaking study, we have identified a critical cutoff value for the CAR score, demonstrating that a CAR value surpassing 0.05 is associated with a decreased completion rate of AC utilizing S-1. This significant finding not only sheds light on the predictive capacity of CAR in anticipating the non-completion of S-1 treatment due to adverse AEs, but also represents a pioneering effort in this domain. The implications of this study are profound, suggesting that CAR could serve as a valuable tool for clinicians in identifying patients who are at heightened risk of encountering challenges in tolerating S-1 treatment, owing to AEs. By leveraging CAR as a predictive metric, healthcare providers can proactively intervene and tailor treatment strategies to mitigate the impact of AEs, thereby optimizing patient outcomes and treatment adherence. Building upon these findings, the research team intends to embark on prospective trials aimed at investigating the efficacy of nutritional interventions among patients exhibiting elevated CAR values. By implementing targeted interventions, the goal is to evaluate the extent to which these interventions can alleviate adverse events and enhance treatment tolerability among the approximately 30% of patients who do not complete S-1 AC treatment. In essence, this study represents a pivotal step forward in personalized cancer care, offering a novel approach to risk stratification and intervention planning. By harnessing the predictive power of CAR and exploring innovative strategies for intervention, clinicians can strive towards improving the delivery of S-1 AC and ultimately enhancing the quality of care for cancer patients.

## Figures and Tables

**Figure 1 cancers-16-00922-f001:**
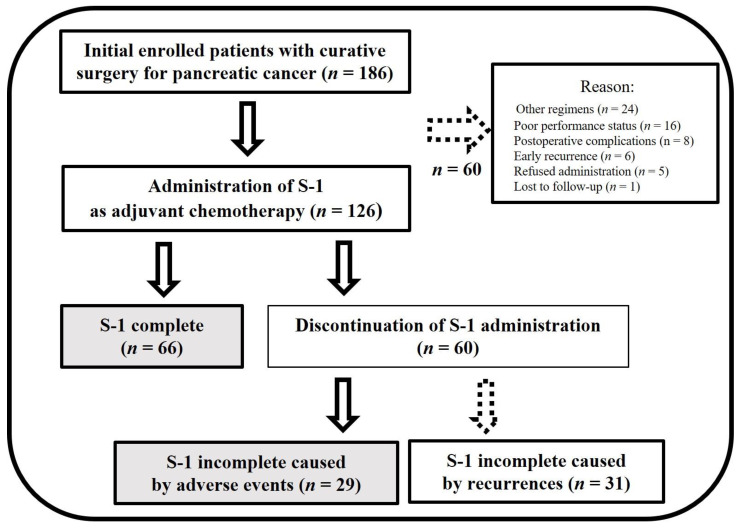
Flowchart of patient selection. Arrows: inclusion criteria, dashed arrows: exclusion criteria, colored box: analyzed patients.

**Figure 2 cancers-16-00922-f002:**
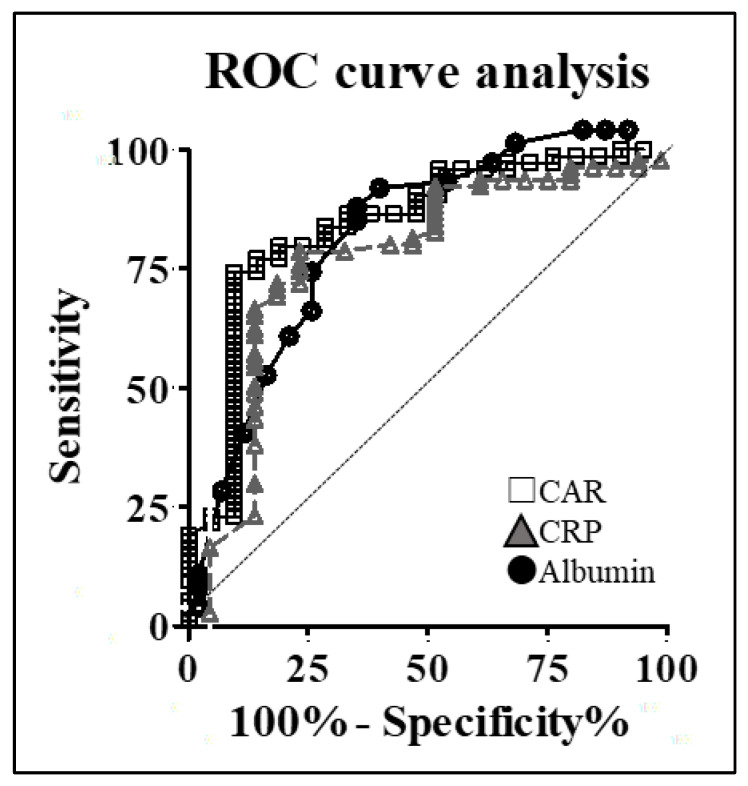
Cutoff selection for C-reactive protein-to-albumin ratio (CAR) using receiver operating characteristic (ROC) curve analysis. Dashed line: Diagonal.

**Figure 3 cancers-16-00922-f003:**
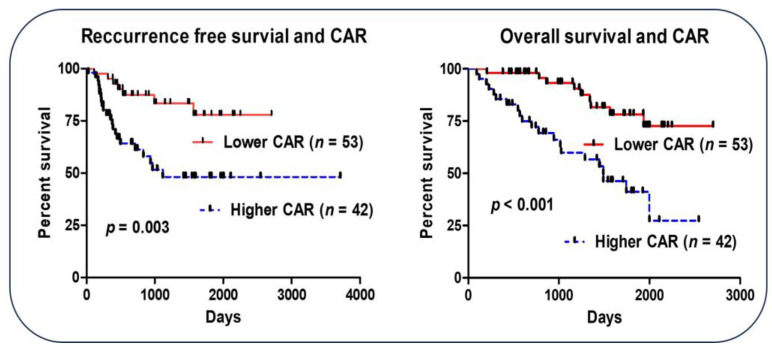
Recurrence-free survival and overall survival according to the CAR cutoff value for pancreatic cancer patients.

**Table 1 cancers-16-00922-t001:** Patient characteristics and perioperative data in the S-1-complete and S-1-incomplete groups.

Patient Characteristics	S-1-Complete Group (*n* = 66)	S-1-Incomplete Group (*n* = 29)	*p* Value
Sex (male)	26 (39.4)	17 (58.6)	0.113
Age (years)	68.8 (47–84)	71.3 (56–85)	0.191
Body mass index (kg/m^2^)	22.6 ± 0.4	21.6 ± 0.6	0.129
ASA-classification			0.054
1 or 2	63 (95.5)	24 (82.8)	
3	3 (4.5)	5 (17.2)	
Neoadjuvant chemotherapy	24 (36.4)	9 (31.0)	0.649
Operation methods			
DP	27 (40.9)	9 (31.0)	
PD	35 (53.0)	14 (65.5)	
TP	4 (6.1)	1 (3.5)	
Operation time (min)	482.1 ± 20.3	533.8 ± 33.6	0.17
Estimated blood loss (mL)	659.6 ± 72.8	883.2 ± 126.	0.116
CD classification grade ≥3	20 (30.3)	5 (17.2)	0.183
Postoperative hospital stay (days)	30.0 ± 2.5	29.9 ± 4.5	0.979
Pathological stage			
1	11 (16.7)	3 (10.3)	
2	53 (80.3)	25 (86.2)	
3	1 (1.5)	0 (0.0)	
4	1 (1.5)	1 (3.4)	

Data are presented as n (%), median [interquartile range], or mean ± standard deviation. DP: distal pancreatectomy; PD: pancreatoduodenectomy; TP: total pancreatectomy.

**Table 2 cancers-16-00922-t002:** Results of blood tests at initiation of AC, perioperative factors, and AE incidence in the S-1-complete and S-1-incomplete groups.

Variables	S-1-Complete Group (*n* = 66)	S-1-Incomplete Group (*n* = 29)	*p* Value
Duration to AC initiation (days)	51.5 ± 4.6	49.3 ± 4.5	0.777
Data at the onset of AC			
Body mass index (kg/m^2^)	21.2 ± 0.3	19.8 ± 0.6	0.026
Alb (mg/dL)	3.8 ± 0.0	3.3 ± 0.1	0.002
CRP (mg/dL)	0.2 ± 0.2	1.0 ± 0.3	<0.001
CEA (ng/mL)	2.5 ± 0.2	2.8 ± 0.4	0.438
CA19-9 (U/mL)	57.8 ± 29.7	45.2 ± 14.2	0.784
CAR	0.04 ± 0.01	0.31 ± 0.10	<0.001
Severe AEs	5 (7.6%)	14 (48.3%)	<0.001
Recurrence after S-1 treatment	29 (43.9%)	17 (58.6%)	0.265

Data are presented as mean ± standard deviation or n (%). AC: adjuvant chemotherapy; Alb: albumin; CRP: C-reactive protein; CEA: carcinoembryonic antigen; CA19-9: carbohydrate antigen 19-9; CAR: CRP-to-albumin ratio; AEs: adverse events.

**Table 3 cancers-16-00922-t003:** Patient characteristics and perioperative factors in the higher-CAR (CAR ≥ 0.05) and lower- CAR (CAR < 0.05) groups.

Variables	CAR < 0.05 (*n* = 53)	CAR ≥ 0.05 (*n* = 42)	*p* Value
Sex (male)	17 (32.1)	26 (61.9)	0.007
Age (years)	69.6 ± 1.2	69.6 ± 1.4	0.937
POCs (CD classification grade ≥3)	10 (18.9)	15 (35.7)	0.064
Data at the onset of AC	25 (46.3)	5 (21.7)	
Alb (mg/dL)	3.8 ± 0.1	3.4 ± 0.1	<0.001
CRP (mg/dL)	0.06 ± 0.01	0.84 ± 0.20	<0.001
CA19-9 (U/mL)	78.3 ± 38.0	24.7 ± 5.6	0.204
AC completion rate (%)	48 (90.6)	18 (42.9)	<0.001
Pathological stage			
1	11 (20.8)	3 (7.1)	
2	41 (77.4)	37 (88.1)	
3	0 (0.0)	1 (2.4)	
4	1 (1.9)	1 (2.4)	
Severe AEs	7 (13.2)	12 (28.6)	0.075

Data are presented as *n* (%) or mean ± standard deviation. AC: adjuvant chemotherapy; POCs: postoperative complications.

**Table 4 cancers-16-00922-t004:** Logistic regression multivariate analysis for S-1 completion.

Variables	Hazard Ratio (95% Confidence Interval)	*p* Value
BMI < 20.0	1.322 (1.041–1.680)	0.022
Albumin < 3.4	3.282 (1.041–1.680)	0.13
CRP > 0.18	1.322 (0.704–15.304)	0.891
Severe AEs	19.897 (3.660–108.157)	<0.001
CAR > 0.05	12.734 (2.064–79.253)	0.006

BMI: body mass index; CRP: C-reactive protein; AEs: adverse events; CAR: CRP-to-albumin ratio.

## Data Availability

The datasets used and analyzed during the current study are available from the corresponding author upon reasonable request.

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
