# Peer review of "C-Reactive Protein-to-Albumin Ratio to Predict Tolerability of S-1 as an Adjuvant Chemotherapy in Pancreatic Cancer"

_cancers, 2024, doi:10.3390/cancers16050922_

Round 1
Reviewer 1 Report
Comments and Suggestions for Authors
Thank you for the opportunity to review this manuscript. This single-institution retrospective study was nicely done and showed the relevance of CAR (exceeding 0.05) in predicting the non-completion of S-1 treatment due to AEs. The study methodology is solid. The conclusion is supported by the data analysis. This manuscript adds value to the literature.
I have the following minor comments:
· List of all abbreviations in the beginning of the main manuscript.
· Table 2. when did you collect the AE data? How could you get severe AE at the onset of AC? Are you referring to POC?
· What is the cutoff for those variables in Table 4?
· Is CAR better than CRP? Figure 2.
· Page 8, the correct statement should be “a CAR cutoff value exceeding 0.05 correlates to lower completion rate of AC with S1”. A CAR cutoff value cannot reduce the completion rate.
Comments on the Quality of English Languagethis manuscript was written well.
Author Response
Reviewer1
- List of all abbreviations in the beginning of the main manuscript.
Thank you for your suggestion. We added the list in our manuscript.
2· Table 2. when did you collect the AE data? How could you get severe AE at the onset of AC? Are you referring to POC?
We apologize for the confusion caused by arranging the frequency of AEs according to the insufficient title and alongside the onset of AC. Thank you very much for bringing it to our attention. We will correct this.
3· What is the cutoff for those variables in Table 4?
The cutoff values are derived from ROC curve analysis for each, so I will add them to the table 4.
4· Is CAR better than CRP? Figure 2.
We will add annotations to ensure clarity regarding which value corresponds to each item.
Reviewer 2 Report
Comments and Suggestions for Authors
The manuscript submitted by Naotake Funamizu and coauthors titled as “C-reactive protein-to-albumin ratio to predict tolerability of S-1 2 as an adjuvant chemotherapy in pancreatic cancer”.
The flow chart in the first looks explains the theme of the manuscript, hence appreciated. But the data regarding S1 Incomplete due to recurrence has not been shared in the upcoming manuscript. The study has been well designed, the number of samples are less and hematological and biochemical parameters were of worth to share.
The study describes AC/ S1 and PC, with the key objective of the study is to evaluate CRP:albumin (CAR), a possible marker for S1 association with non completion due to AEs with PC patients.
Initially N= enrolled, N=60 patients excluded (unable to initiate S-1 AC), N=126 were part of the study, further N=66 completed, whereas N=60 withdrawn being AS-AEs(N=29) & AS-Recur ( N=31).
It is highly recommended that first table should contains data of S1 complete, S1 incomplete (both adverse events and recurrence)
Referring to line 87, hematological and biochemical analysis are recommended to be part of data presented in the manuscript or as supplementary data with statistical analysis.
Refer to Table 1 and 2, contains S-1 Patient characteristics and perioperative data in the S-1-complete and S-1-incomplete groups. AS-AE(N=29), why AS-Recur (N=31) the data of has not been shown, it recommended that it should be part of supplementary data. It will be interesting to see the statistical analysis of three groups.
Because the finding of Alb, CRP & CAR are very interesting and further endorsed by AEs, could the author share the finding of Alb, CRP & CAR endorsed by AS-Recurrence (N=31).
In addition to that what are possible explanations for decrease of BMI S1-AEs? Is it the trend is similar in case of S1-AERecur (N= 31)?
Refer to the statements in manuscript” An observed CAR value of 0.05 or greater was linked to an increased risk of S-1 therapy incompletion” is the statement is only S1AE ( N=29) or can be applied to S1Recurrance (N=31), if it can not be applied what may be the possible explanation?
Recent references are highly appreciated such as 2023.
Author Response
Reviewre2
- It is highly recommended that first table should contains data of S1 complete, S1 incomplete (both adverse events and recurrence)
Thank you for your valuable advice. We added the data as you suggested. But we excluded rec 31 patients from S1 incomplete group. So, the data shows the number of rec patients in incomplete group (29 patients). Moreover, table 1 shows patients' backgrounds before AC. That is why we added the data in table2.
- Referring to line 87, hematological and biochemical analysis are recommended to be part of data presented in the manuscript or as supplementary data with statistical analysis.
We are currently collaborating with other institutions on a joint study using these data. We would like to provide the data to interested parties only upon request. Your understanding would be greatly appreciated.
- Refer to Table 1 and 2, contains S-1 Patient characteristics and perioperative data in the S-1-complete and S-1-incomplete groups. AS-AE(N=29), why AS-Recur (N=31) the data of has not been shown, it recommended that it should be part of supplementary data. It will be interesting to see the statistical analysis of three groups .Because the finding of Alb, CRP & CAR are very interesting and further endorsed by AEs, could the author share the finding of Alb, CRP & CAR endorsed by AS-Recurrence (N=31).
Thank you for your precious comment.
We had sufficient discussion repeatedly regarding whether to include the data from the 31 excluded patients in the analysis. Our focus remains solely on whether AE-based S-1 treatment dropout can be predicted through nutritional assessment using CAR ratio. Therefore, patients who had their regimen changed for treatment purposes were not considered dropouts; hence, they were excluded. With the deadline for responses to the review being only 5 days, and considering that the data of the excluded patients have already been discarded, we do not have their data in the database for this study. If an extension is granted, it would be possible to include supplementary data such as the data for the 31 patients with CAR in the supplement.
- In addition to that what are possible explanations for decrease of BMI S1-AEs? Is it the trend is similar in case of S1-AERecur (N= 31)?
As stated in the text, weight loss in gastric cancer patients has been reported to hinder the continuation of S1. In this study, low BMI also showed a similar trend. Therefore, administering S-1 without sufficient oral intake postoperatively is suggested to make continuation difficult. Conversely, it may be better to start S-1 when the weight has improved to some extent. In contrast, excluded patients were not investigated in this study.
- Refer to the statements in manuscript” An observed CAR value of 0.05 or greater was linked to an increased risk of S-1 therapy incompletion” is the statement is only S1AE ( N=29) or can be applied to S1Recurrance (N=31), if it can not be applied what may be the possible explanation?
First, we appreciate your insightful comments. We think that the speed of recurrence is associated with disease conditions. So, we believe predicting the progression of the disease based on nutritional status is difficult. Therefore, we excluded the 31 individuals who experienced recurrence in order to change the treatment regimen. However, they were not dropouts due to AEs. That is why our conclusion stated that “This study is the first to demonstrate the relevance of CAR in predicting the non-completion of S-1 treatment due to AEs.”
- Recent references are highly appreciated such as 2023.
We changed the reference (cancer statistics 2023).
- Page 8, the correct statement should be “a CAR cutoff value exceeding 0.05 correlates to lower completion rate of AC with S1”. A CAR cutoff value cannot reduce the completion rate.
Thank you for your guidance. We will revise the sentence as you suggested.